



# Exploring the influence of spatio-temporal scale differences in Coupled Data Assimilation

Lilian Garcia-Oliva[1], Alberto Carrassi[2], and François Counillon[1, 3]

[1]Geophysical Institute, University of Bergen and Bjerknes Centre for Climate Research, Bergen, Norway
[2]Department of Physics and Astronomy "Augusto Righi", University of Bologna, Italy
[3]Nansen Environmental and Remote Sensing Center and Bjerknes Centre for Climate Research, Bergen, Norway

**Correspondence:** Lilian Garcia-Oliva (lilian.garcia@uib.no)

**Abstract.** Identifying the optimal strategy for initializing coupled climate prediction systems is challenging due to the spatio-temporal scale separation and disparities in the observational network. We aim to clarify when strongly coupled data assimilation (SCDA) is preferable to weakly coupled data assimilation (WCDA). We use a two-components coupled Lorenz-63 system and the Ensemble Kalman Filter (EnKF) to compare WCDA and SCDA for diverse spatio-temporal scale separations and observational networks — only in the atmosphere, the ocean, or both components. When both components are observed, SCDA and WCDA yield similar performances. However, sometimes SCDA performs marginally worse due to its higher sensitivity (as opposed to WCDA) to key approximations in the EnKF — linear analysis update and sampling error. When observations are only in one of the components, SCDA systematically outperforms WCDA. The spatio-temporal scale separation determines SCDA's performance in this scenario, and the largest improvements are found when the observed component has a smaller spatial scale. This suggests that SCDA of fast atmospheric observations can potentially improve the large-slow ocean component. Conversely, observations of the fine ocean can improve the large atmosphere at a comparable temporal scale. However, when both components are highly chaotic, and the observed component's spatial scale is the largest, SCDA does not improve over WCDA. In such a case, the cross-updates may become too sensitive to data assimilation approximations.

## 1 Introduction

Environmental and climate prediction systems nowadays are transitioning toward the use of coupled models — from numerical weather prediction (NWP) to seasonal-to-decadal (S2D) climate predictions — with a target to build seamless forecasting systems that can perform across various timescales (Shukla, 2009). In general, coupled prediction systems to date assimilate data independently in each of the different sub-components (Meehl et al., 2021): e.g., atmospheric observations are used to infer the state of the atmosphere, ocean observations are used for the ocean, and so on. However, this technologically convenient strategy results in the suboptimal use of observations across the different components and can degrade the dynamical





consistency of the system and generate spurious drifts (Penny and Hamill, 2017). A key challenge of assimilating observations across components is the spatio-temporal scale separation of the climate system.

The climate system features a wide range of temporal and spatial scales that go beyond the stereotypical association of fast
and large for the atmosphere and slow and small for the ocean. For instance, the time and spatial scale of the ocean and the atmosphere are of the same order in the equatorial Pacific, dominated by the El Niño-Southern Oscillation (ENSO). In the North Atlantic, the fast North Atlantic Oscillation (NAO) strongly influences the slower and larger Atlantic meridional overturning circulation and the Atlantic multidecadal variability (Clement et al., 2015), but with evidence of a feedback mechanism of the slow ocean variability to NAO (Zhang et al., 2019). Quite similarly, in the Pacific, the fast and local Aleutian Low variability
is a primary driver of the variability of the Pacific decadal variability (PDV, being slow and large scale). The Aleutian Low is influenced by both the local wind variability and ENSO, and a feedback mechanism of the PDV on the Aleutian Low has been proposed via Rossby waves influencing the position of the Kuroshio (i.e., small-scale ocean front Newman et al., 2016).

Data assimilation (DA) methods estimate the state of a dynamical system based on observations, a dynamical model, and statistical information on the error terms (Carrassi et al., 2018). Coupled DA (CDA) is produced with fully coupled models and
aims at providing balanced and self-consistent states within the coupled model (Zhang et al., 2020). CDA is executed in either weakly or strongly coupled fashion (with acronyms WDCA and SCDA, respectively; Laloyaux et al., 2016). In WCDA, the assimilation is applied to the individual components separately by using the observations available for that component. Notably, the observations can still impact across components via the dynamical coupling between the assimilation cycle, unlike with uncoupled data assimilation (i.e., performed with an uncoupled model). On the other hand, in SDCA, the observations from
one component impact the other components directly during the assimilation. SCDA is, in principle, the best approach for CDA since the statistical and dynamical assimilation of observations through the coupled cross-covariance potentially provides more information and produces better and more dynamically balanced analysis (Penny and Hamill, 2017).

Comparison of SCDA and WCDA has been studied with dynamical systems of increasing complexity (Liu et al., 2013; Sluka et al., 2016; Penny et al., 2019; Tondeur et al., 2020). While SCDA gives some clear improvements over WCDA in some
locations, configurations (e.g., observational network, toy models) and selected processes (Sandery et al., 2020; Kalnay et al., 2023; Sun et al., 2020) degradations are also found. Such degradations have been attributed to the interconnection of processes with disparate spatio-temporal scales in the climate system, approximations in DA, and model error. For instance, model error and limited ensemble sizes can hinder the accurate estimation of the system's coupled cross-correlation (Han et al., 2013; Tardif et al., 2014; Lu et al., 2015a). Furthermore, SCDA improves results if observations are only found in one component
(Sluka et al., 2016), but conclusions are often not as clear when both components are partially observed (Sluka, 2018). This typically motivates the use of ad-hoc methods such as cross-component localization (Frolov et al., 2016; Stanley et al., 2024), or the use of time average covariance, like the Leading Average Coupled Covariance (Lu et al., 2015a, b; Sun et al., 2020, LACC method) to circumvent these limitations.

This study aims to further clarify the conditions and framework favouring one CDA approach over the other, attempting to
discern the key model and observation features driving the results. We use a low-order coupled system and extensively compare the different approaches for a wide range of temporal and spatial scales and observation configurations. This can help us to





anticipate when the SCDA is expected to outperform WCDA in an operational configuration, thus legitimizing the allocation of resources to migrate from WCDA to SCDA.

The paper is structured as follows. In Sect. 2, we describe the coupled model used in this paper. We identify how the
combination of spatio-temporal scale differences between the model's components impacts the sensitivity to initial conditions and the general characteristics of the system. In Sect. 3, we describe the experimental setup, the metrics used, and the set of DA experiments performed. The results of our experiments are presented and discussed in Sect. 4. We close this paper with our concluding remarks and outlook in Sect. 5.

## 2   Lorenz multiscale coupled system

We use two coupled Lorenz (Lorenz, 1963, L63) models, as introduced by Peña and Kalnay (2004), as a proxy of real complex coupled multiscale dynamical systems (e.g., atmosphere-ocean). The system consists of a fast and a slow component, with the following equations:

$$
\begin{aligned}
\dot{x} &= \sigma(y - x) - c(SX + k) \\
\dot{y} &= rx - y - xz + c(SY + k) \\
\dot{z} &= xy - bz \\
\dot{X} &= \tau\sigma(Y - X) - c(x + k) \\
\dot{Y} &= \tau r X - \tau Y - \tau S X Z + c(y + k) \\
\dot{Z} &= \tau S X Y - \tau b Z.
\end{aligned}
\tag{1}
$$

The low-case variables $(x, y, z)$ indicate the fast component (considered in the following to be our atmosphere), while the
capital variables $(X, Y, Z)$ indicate the slow component (in the following to be the ocean); and $\dot{x}$ indicates the derivative of $x$ with respect to time. The parameters $\sigma$, $b$, and $r$ (Table 1) are set to the default values of the L63 (Lorenz, 1963). The coupling strength is modulated by the parameter $c$, here set to the value $c = 0.15$. The coupling occurs only through the $(x, y) \leftrightarrow (X, Y)$ components. The parameters $S$ and $\tau$ control the components' spatial and temporal scale differences. In Peña and Kalnay (2004), $S = 1.$ and $\tau = 0.1$, meaning that both components have approximately the same spatial scale, but the
slow component is ten times slower. The effective spatial scale difference also depends on the values of the other parameters (even from the nonlinear interactions coming from the model) but is predominantly sensitive to $S$ (Sect. 2.1). Finally, $k = 10$ is the uncentering parameter that shifts the phase of each component during the coupling. Since the coupling is relatively weak, the "slow→fast" information is rapidly dissipated, while the "fast→slow" interaction introduces a "weather noise"-like signal to the slow component (Peña and Kalnay, 2004). With these parameter values, the system mimics a weak extratropical
ocean-atmosphere coupling.

We integrate the system using the fourth-order Runge-Kutta numerical scheme, using an adimensional time step $\mathrm{d}t$ of $10^{-2}$ time units (TU) and initial condition $\mathbf{x}_0^{\mathrm{T}} = (x_0, y_0, z_0, X_0, Y_0, Z_0)^{\mathrm{T}} = (0., 1., 0., 0., 0., 0.)^{\mathrm{T}}$. We integrated the system over 1500





**Table 1.** Set of parameters for the coupled L63, Eq. (1).

| $\sigma$ | $b$ | $r$ | $c$ | $S$ | $\tau$ | $k$ |
|------|-----|-----|------|-----|------|-----|
| 10. | 8/3 | 28 | 0.15 | 1. | 0.1 | 10. |

**Timeseries of the copupled system $(S, \tau)$= (1., 0.1)**

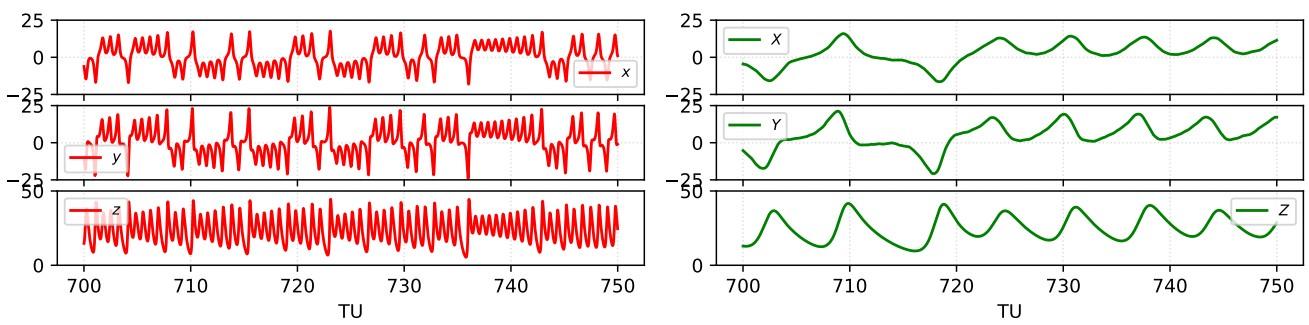

**Figure 1.** Time series of the atmospheric (left) and ocean (right) variables. The model uses here the configuration on Table 1, of the extratropical atmosphere-ocean system.

TU and discarded the initial transient period of approximately 40 TU for our following analyses. Figures 1 and 2 show the time series and a 2-dimensional projection of the system's attractor using the default parameters as in Peña and Kalnay (2004)

(Table 1). Figure 1 shows the time scale difference between the fast and slow components while they have similar amplitude. Figure 2 displays the projections on the $(x,y)$ and $(X, Z)$ planes of the system's attractor: the atmospheric and ocean portion of the attractor share the same topological shape, but the atmosphere has a much higher frequency.

**Copupled system attractor $(S, \tau)$= (1., 0.1)**

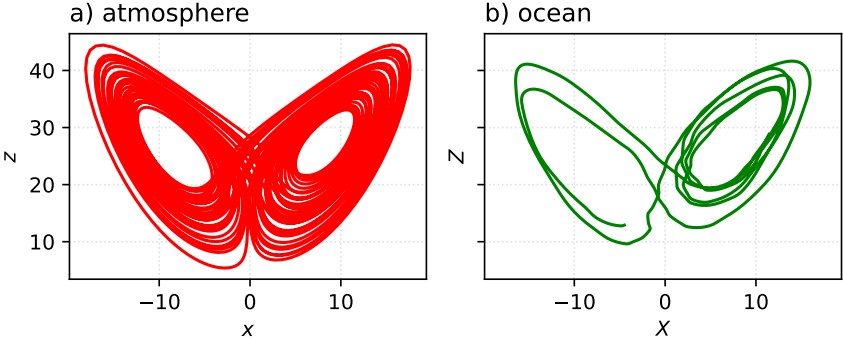

**Figure 2.** Standard configuration attractors of the extratropical atmosphere-ocean system during the 700–750 TU.





We commence our analysis by exploring the impact of $S$ and $\tau$, the parameters controlling the amplitude and time-scale mismatch between atmosphere and ocean, on the physical and dynamical properties of the system. In particular, we focus on (1) the energy partition between components, the effective temporal separation, and the instantaneous cross-component covariance (Sect. 2.1); (2) the error propagation by computing the spectrum of Lyapunov exponents (LEs), the Kolmogorov entropy (KE), the Kaplan-Yorke attractor dimension (KY-dim), and the flow's divergence ($\nabla \cdot \mathbf{f}$) (Sect. 2.2); and (3) the error propagation across components (Sect. 2.3).

## 2.1 Parameters influence on energy, time-scale separation and cross covariance

To understand how the effective difference in temporal (or spatial) scales between the components is determined by $S$ and $\tau$, we compute the energy $E_a/E_o$ and period $T_a/T_o$ ratio with varying values of $S$ and $\tau$. The sub-index $a$ denotes the atmosphere and $o$ the ocean components.

The energy $E$ of the two components ($E_o$ for $X, Y, Z$ and $E_a$ for $x, y, z$) of the coupled system are computed as follows:

$$E = \sum_{n=1}^{N} x_n^2 + \sum_{n=1}^{N} y_n^2 + \sum_{n=1}^{N} z_n^2 \tag{2}$$

where $N$ is integration length, and $x_n$, $y_n$, and $z_n$ are the component's variables at time $n$.

We estimate the component's period $T$, i.e. the dominant time scale, as the period at which the power spectrum density (PSD) reaches its maximum. To estimate the PSD, we use the component's magnitude $m$:

$$m = \|x\| = (|x|^2 + |y|^2 + |z|^2)^{1/2}. \tag{3}$$

The sensitivity of energy $E_a/E_o$ and period $T_a/T_o$ ratios to $S$ and $\tau$ is explored in Fig. 3. The relative energy content of each component (Fig. 3a) shows that the energy of the ocean ($E_o$) is mostly inversely proportional to $S$ and that the temporal scale has only a little influence on it. As it could have been anticipated, the temporal separation is uniquely sensitive to $\tau$. Figure 3 demonstrates that one can change these two parameters separately to modulate the spatial or temporal scale differences accordingly.

We analyze the dependence of information transfer between components on the spatio-temporal parameters by computing the cross-component instantaneous correlation of the coupled system $\mathbf{C}$ over 500 TU for all $(S, \tau)$ configurations and calculate the spectral norm of it (Fig. 4). We consider the sub-matrix $\mathbf{C}^{cc} = c_{ij}$, where $i = 1, 2, 3$ and $j = 4, 5, 6$. The spectral norm of a matrix $\mathbf{A}$ is computed as follows:

$$\|\mathbf{A}\|_2 = \rho(\mathbf{A}^* \mathbf{A})^{1/2}, \tag{4}$$

where $\rho(\mathbf{A}^* \mathbf{A}) = \max_{1 \leq i \leq n} |\lambda_i|$ is the spectral radius of $\mathbf{A}$, the maximum modulus of the $n$ eigenvalues $\lambda_i$ of $\mathbf{A}$; and $\mathbf{A}^*$ is the conjugate of $\mathbf{A}$. Since the correlation matrix $\mathbf{C} \in \mathbb{R}^{n \times n}$ then $\mathbf{C}^* = \mathbf{C}^{\mathrm{T}}$.



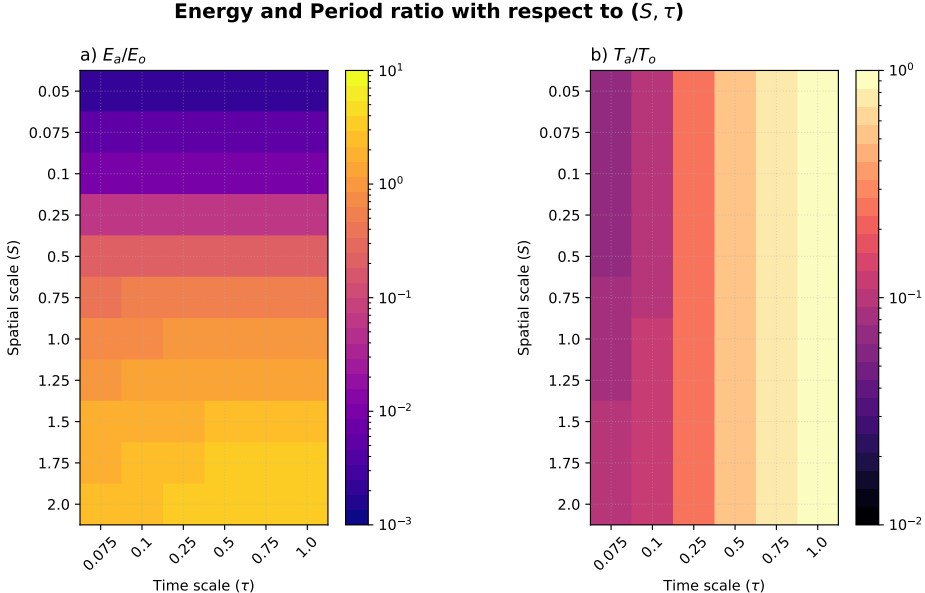

**Figure 3.** Dependence of the a) energy ratio $E_a/E_o$ and b) period ratio $T_a/T_o$ on $S$ and $\tau$ with a logarithmic colourbar. Note that the ocean's spatial scale $S$ (y-axis) increases upward.

The pattern in Fig. 4 reveals that the information flow across the system's components is heavily influenced by $S$ and $\tau$. The cross-component correlation decreases as the spatial scale difference increases. However, and somehow unexpected, the cross-component correlation increases as the time scale difference increases. The cross-covariance shows a maximum when the ocean component $(X, Y, Z)$ has a smaller spatial scale and slower time scale than the atmospheric component $(x, y, z)$, implying a high information flow. Conversely, a large and fast ocean hampers the flow of information. These findings align with what was found by Tondeur et al. (2020) in relation to time-scale difference only.

## 2.2 Influence of the spatio-temporal scale mismatch on the chaotic behaviour of the coupled system

Here, we assess how the spatio-temporal parameters influence the chaoticity of the coupled system. We quantify the sensitivity of the system to initial conditions by exploring its Lyapunov spectrum. We use the Benettin algorithm (Benettin et al., 1980), as described by Ayers et al. (2023), to compute the Lyapunov exponents (LEs) and the first backward Lyapunov vector (LV1). The computation of LEs and LV1 is performed over an integration of $1 \times 10^4$ TU, thus ensuring convergence of the method. To understand the effect of the coupling on the dynamics of the coupled system, we first compare the coupled L63 system (Eq. (1)) to two uncoupled L63 systems (Eq. (5) Lorenz, 1963).



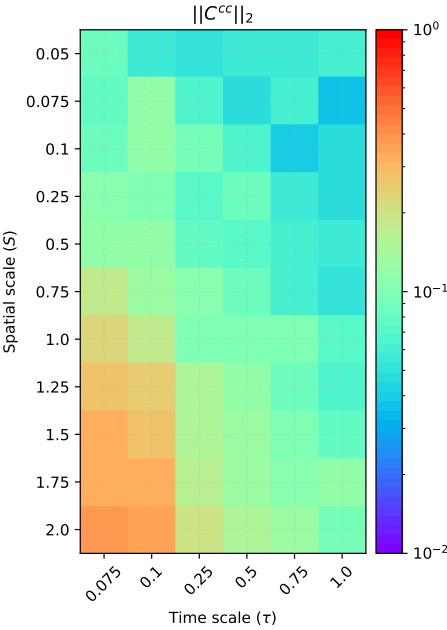

**Figure 4.** Instantaneous cross-covariance spectral norm $||\mathbf{C}^{cc}||_2$ for each $(S, \tau)$ combination. The value presented is the average of 25 runs using different initial conditions. Colourbar is in logarithmic scale

$$\dot{x} = \tau\sigma(y - x)$$
$$\dot{y} = \tau(rx - y - Sxz) \qquad\qquad (5)$$
$$\dot{z} = \tau(Sxy - bz).$$

For the uncoupled atmosphere, we set $(S = 1., \tau = 1.)$, for the ocean $(S = 1., \tau = 0.1)$, and compare them with the coupled system (Eq. (1)) with the parameterization in Table 1. Figure 5 shows the Lyapunov spectrum of the uncoupled sub-components (atmosphere in red; ocean in green) and the coupled system (blue). Both uncoupled systems possess one positive ($\lambda_i^+$), one negative ($\lambda_i^-$), and one neutral ($\lambda_i^0$) LE. However, while the atmosphere's LEs are substantially different, in the ocean, they are much closer to each other. The ocean appears only very marginally unstable, with the largest LE just above zero. Thus,

the uncoupled-unforced ocean is nearly stable. This is also confirmed by the Kolmogorov-Sinai entropy and the divergence (KS-E and $\nabla \cdot \mathbf{f}$, respectively, in Table 2), which are around one order of magnitude smaller than for the atmosphere. Notably, although the ocean is much stabler than the atmosphere, their attractors' dimensions (measured by the Kaplan-York dimension KY-dim, in Table 2) are the same.

    The Lyapunov spectrum of the coupled system is a mix of the uncoupled atmosphere and ocean spectra. The coupling does

not affect the number of positive Lyapunov exponents, which remains one. Instead, it introduces two near-neutral exponents while retaining the two negative exponents from each uncoupled system. The presence of these additional exponents with



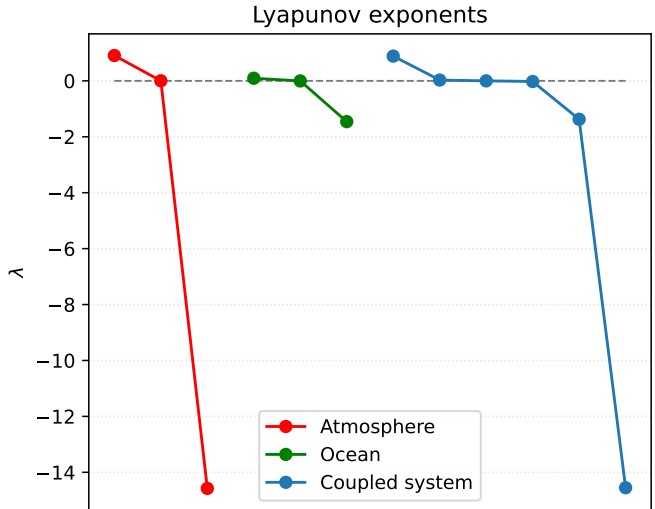

**Figure 5.** Lyapunov exponents (LEs) for the uncoupled atmosphere (L63) in red ($S = 1, \tau = 1$), in green the uncoupled ocean (L63-like, with $S = 1, \tau = 0.1$) and in blue the coupled system ($S = 1, \tau = 0.1$).

values close to zero is commonly referred to as *quasi-degeneracy*. The emergence of these quasi-neutral modes has been found to be related to the coupling itself (Penny et al., 2019; Tondeur et al., 2020) in models of higher complexity, such as MAOOAM (Cruz et al., 2016). Moreover, their presence has tremendous implications when designing efficient DA methods to control error

growth (Carrassi et al., 2022).

**Table 2.** Stability analysis of the coupled and uncoupled system.

|  | Atm | Ocn | Coupled system |
|---|---|---|---|
| KY-dim | 2.062 | 2.061 | 4.646 |
| KS-E | 0.904 | 0.089 | 0.898 |
| $\nabla \cdot \mathbf{f}$ | -13.667 | -1.367 | -15.033 |
| $\lambda_i^+$ | 0.906 | 0.094 | 0.885 |
| $\lambda_i^0$ | 0.002 | -0.002 | 0.029 |
|  |  |  | 0.0003 |
|  |  |  | -0.022 |
| $\lambda_i^-$ | -14.57 | -1.457 | -1.373 |
|  |  |  | -14.55 |

We now investigate in Fig. 6 how the chaotic behavior of the coupled system changes with $(\tau, S)$ parameters. From the system's Jacobian (Eq. (7)), we can already anticipate that $\tau$ will play a larger role in modulating the system's degree of chaos





than $S$. However, since $S$ appears in the cross-component terms (from the ocean to the atmosphere, Eq. (7)), it can potentially influence the dynamical properties of the system, regardless of the coupling parameter $c$ (Sect. 2.3).

Changing either $\tau$ or $S$ does not alter the shape of the Lyapunov spectrum — i.e., the number of positive, near-neutral, neutral, and negative LEs (not shown). However, changing $\tau$ impacts the degree of chaos of the system and the attractor's dimension (KS-E, KY-dim and $\nabla \cdot \mathbf{f}$ in Fig. 6), independently of the value of $S$. As the time scale separation decreases ($\tau \to 1$), the system becomes more unstable, and the dimension of the attractor decreases. However, when $\tau$ gets too large, the KY-dim saturates at 4. Since the KY-dim approaches the phase space dimension ($n = 6$), the entropy decreases (near zero values), and

the divergence increases; we have that the dynamics become similar to a Hamiltonian system (it approximates a conservative system) as the ocean becomes slower and that the error will evolve along the complete phase space.

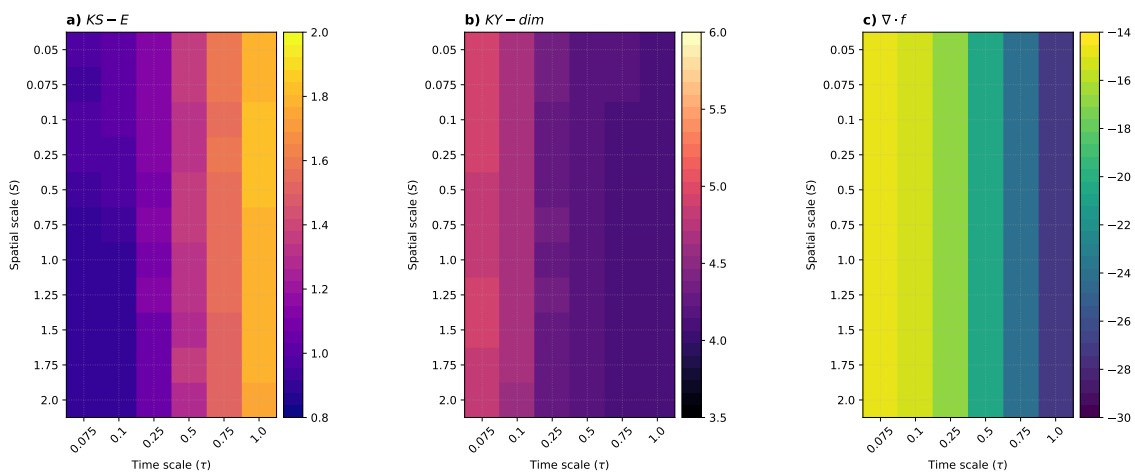

**Figure 6.** Stability analysis of the coupled system in dependence of the spatio-temporal parameters $S$ and $\tau$ a) Kolmogorov-Sinai entropy *KS-E*, b) Kaplan-York dimension *KY-dim*, and c) divergence of the flow $\nabla \cdot \mathbf{f}$. Note that each quantity has its own limits, and the colourbar is on a linear scale.

    We finally analyze how the projection of LV1 on the state vector varies with the spatio-temporal parameters, which helps to understand or discriminate the source of instabilities in the state vector. To this end, we show the ratio between the projection on the atmosphere ($\text{LV1}_a$) and that of the ocean ($\text{LV1}_o$) component in Fig. 7. The LV1 projection on the atmospheric variables is

generally larger, implying that the dominant source of error propagation relies on the atmosphere. This behaviour is independent of the spatial scale of the ocean ($S$) as long as the parameter $\tau$ is less than one (ocean slower than the atmosphere). When both components have the same time scale ($\tau = 1$), the LV1 projection is larger in the component with the largest spatial scale, i.e., the projection on the ocean component decreases when $S > 1$. When $\tau > 0.25$, the error propagation is relatively even on both components. However, when $\tau < 0.25$, the error propagation towards the ocean is almost two orders of magnitude smaller than

towards the atmosphere.





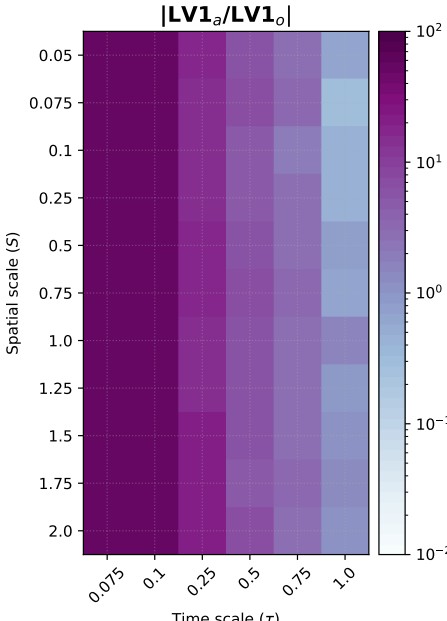

**Figure 7.** Ratio of the normalized LV1 over atmosphere and ocean. Note that the colourbar is on a logarithmic scale.

## 2.3 Error propagation across model's components

In this section, we use the (linearised) dynamical arguments from Tondeur et al. (2020) to investigate the error propagation from one component to the other, and how it depends on the spatio-temporal scale separation parameters $S$ and $\tau$. In this context the evolution of a perturbation $\boldsymbol{\xi}(t)$ in the system is governed by the tangent linear system:

$\dot{\boldsymbol{\xi}}(t) = \mathbf{J}\,\boldsymbol{\xi}(t)$ (6)

where $\mathbf{J}$ is the Jacobian of such system. For our coupled system, in Eq. (1), the Jacobian reads:

$$
\mathbf{J} = \begin{bmatrix}
-\sigma & \sigma & 0 & -cS & 0 & 0 \\
r-z & -1 & -x & 0 & cS & 0 \\
y & x & -b & 0 & 0 & 0 \\
-c & 0 & 0 & -\tau\sigma & \tau\sigma & 0 \\
0 & c & 0 & \tau(r-SZ) & -\tau & -\tau SX \\
0 & 0 & 0 & \tau SY & \tau SX & -\tau b
\end{bmatrix}.
$$ (7)

Let write our coupled system in Eq. (1) in the following general form:





$$\begin{aligned}
\dot{\mathbf{x}} &= \quad \boldsymbol{f}(\mathbf{x}, \mathbf{X}), \\
\dot{\mathbf{X}} &= \quad \tau \boldsymbol{g}(\mathbf{x}, \mathbf{X}),
\end{aligned} \tag{8}$$

where $\mathbf{x}$ and $\mathbf{X}$ indicate the atmosphere and ocean state respectively, and $\tau$ is the time scale. Therefore, over the interval $\mathrm{d}t$, the forecast error in the atmosphere $\boldsymbol{\zeta}(t)$ and in the ocean $\boldsymbol{\eta}(t)$ evolve according to:

$$\begin{bmatrix} \dot{\boldsymbol{\zeta}}(t) \\ \dot{\boldsymbol{\eta}}(t) \end{bmatrix} = \begin{bmatrix} \mathbf{F}_a & \mathbf{F}_o \\ \tau \mathbf{G}_a & \tau \mathbf{G}_o \end{bmatrix} \begin{bmatrix} \boldsymbol{\zeta}(t) \\ \boldsymbol{\eta}(t) \end{bmatrix}. \tag{9}$$

After re-arranging the terms in Eq. (7), we can identify the form of the cross-component error propagation terms, $\mathbf{F}_o$ and $\mathbf{G}_a$ as:

$$\mathbf{F}_o = \frac{\partial \boldsymbol{f}}{\partial \mathbf{X}} = \begin{bmatrix} -cS & 0 & 0 \\ 0 & cS & 0 \\ 0 & 0 & 0 \end{bmatrix} \tag{10}$$

and

$$\mathbf{G}_a = \frac{\partial \boldsymbol{g}}{\partial \mathbf{x}} = \begin{bmatrix} -c/\tau & 0 & 0 \\ 0 & c/\tau & 0 \\ 0 & 0 & 0 \end{bmatrix}. \tag{11}$$

In these expressions, $\mathbf{F}_o$ represents the *ocean $\rightarrow$ atmosphere*, while $\mathbf{G}_a$ the *atmosphere $\rightarrow$ ocean* propagation of error. We can see that $\mathbf{F}_o$ depends explicitly on the spatial parameter $S$. Thus, the error propagation from the slow (ocean) to the fast

(atmosphere) component depends exclusively on $S$ and increases as $S$ does. On the other hand, $\mathbf{G}_a$ depends only on the temporal scale $\tau$, and it is inversely proportional to it. Therefore, the error propagation from the fast to slow components increases as the parameter $\tau$ decreases. Using this information, we can plot the competing direction of error propagation, which we can take as the ratio of the norm of $\mathbf{G}_a$ and $\mathbf{F}_o$. For this, we use the Frobenius norm, defined for a matrix $\mathbf{A}$ as $\|\mathbf{A}\| = [\mathrm{tr}(\mathbf{A}^*\mathbf{A})]^{1/2}$. Thus, the competing direction of error porpagation is $\|\mathbf{G}_a\|\|\mathbf{F}_o\|^{-1} = (S\tau)^{-1}$; we illustrate this

dependence in Fig. 8.

Figure 8 elucidates the role of $S$ and $\tau$ on the error propagation across components. The figure has three separate regions, in which $\|\mathbf{G}_a\| > \|\mathbf{F}_o\|$, $\|\mathbf{G}_a\| \approx \|\mathbf{F}_o\|$, and $\|\mathbf{G}_a\| < \|\mathbf{F}_o\|$. Thus, when the temporal scale separation vanishes and the ocean's spatial scale increases ($\tau^{-1} > S$), the error propagation is dominated by the *atmosphere $\rightarrow$ ocean* term $\mathbf{G}_a$ (blue region in Fig. 8). In the opposite case, when the temporal scale separation increases and the ocean's spatial scale decreases ($\tau^{-1} < S$),

the error mostly propagates from the *ocean $\rightarrow$ atmosphere* term $\mathbf{F}_o$ (red region in Fig. 8).




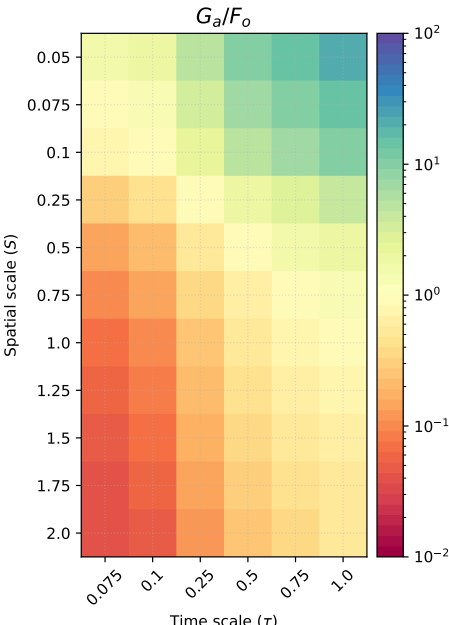

**Figure 8.** Competing direction of error propagation $\|\mathbf{G}_a\|\|\mathbf{F}_o\|^{-1}$ as function of the spatio-temporal scale separation $(S, \tau)$. The blue region indicates cases where the impact of *atmosphere → ocean* is larger. The red region is where the *ocean → atmosphere* error propagation is dominant. Note that the colourbar is in logarithmic scale and is centred around 1.

## 3   Data Assimilation experiments

### 3.1   Experimental Setup

We conduct a set of DA experiments with the coupled L63 for different values of parameters $S$ and $\tau$ to reflect spatio-temporal separations between the two components of the system in Eq. (1). We used the stochastic Ensemble Kalman Filter (EnKF
Evensen, 2003) and compared weakly and strongly coupled data assimilation (WCDA and SCDA, respectively).

We use an idealized perfect twin experiment framework (Arnold and Dey, 1986), i.e., the model used for generating synthetic observations is the same as that used for DA. The synthetic observations are generated by adding zero-mean Gaussian noise to a reference simulation (hereafter referred to as True). Here, we only observed $(y, Y)$ variables. This choice follows what was done by, e.g., Yoshida and Kalnay (2018) and Quinn et al. (2020) that showed these two variables to be more informative in
the Lorenz model (Yang et al., 2006). Similar performances were found when observing the full system (not shown).

The observation error standard deviation $\sigma$ is equal to 2.5% of the system's natural variability (i.e., the time-wise standard deviation of each model variable). This implies that the observation error covariance matrix $\mathbf{R}$ depends on the spatio-temporal scales of each component (see Sect. 2.1). The observational error is uncorrelated; therefore, $\mathbf{R}$ is diagonal with the observational error variance along the diagonal.



The DA is performed using an observational interval equal to one-fifth of the error-doubling time of an uncoupled L63 system (see Sect. 3.4 for further details for this choice). To prevent filter divergence, we used an adaptive inflation scheme (Sect. 3.3) and 20 ensemble members. We run the model for 850 TU, allowing for 150 TU for spin-up before the start of the DA. We use such a long spin-up time to allow all the system's $(S, \tau)$ combinations to evolve beyond the transient period (especially for the cases with a very slow ocean). The error statistics (Sect. 3.5) are computed over the last 600 TU in a similar

way to the experiments of Yoshida and Kalnay (2018) and Quinn et al. (2020). We also repeat the DA experiments 30 times with different initial conditions $\mathbf{x}_0^{\mathrm{T}}$ and different observation perturbations to assess the system's performance robustly.

### 3.2 Data assimilation with the Ensemble Kalman Filter EnKF

The Ensemble Kalman Filter (EnKF, Evensen, 2003) is a Monte Carlo-like sequential DA methodology consisting of a forecast step alternated with an update phase (analysis). During the first phase, the ensemble of states (the ensemble) is integrated

forward in time (forecast) from the previous ensemble of analysis states. During the second phase, observations are used to update (analyze) the ensemble for the next iteration. The method uses ensemble covariance to provide flow-dependent correction and performs a linear analysis update.

We denote the ensemble forecast $\mathbf{X}^{\mathrm{f}} \in \mathbb{R}^{n \times N}$. The superscript $f$ stands for forecast, $N$ is the ensemble size, and $n$ is the dimension of the state. The model error is assumed to follow a Gaussian distribution with zero mean. The ensemble mean

is denoted $\overline{\mathbf{x}}^{\mathrm{f}}$ and the ensemble anomalies are $\mathbf{A}^{\mathrm{f}} = \mathbf{X}^{\mathrm{f}} - \overline{\mathbf{x}}^{\mathrm{f}} \mathbf{1}^{\mathrm{T}}$, where $\mathbf{1} \in \mathbb{R}^{N \times 1}$ has all its values equal to 1. Under the aforementioned hypothesis, the ensemble covariance $\mathbf{P}^{\mathrm{f}}$ is an approximation of the true forecast error, $\epsilon$, covariance matrix:

$$\overline{\epsilon \epsilon^{\mathrm{T}}} \approx \mathbf{P}^{\mathrm{f}} = (N-1)^{-1} \mathbf{A}^{\mathrm{f}} \mathbf{A}^{\mathrm{fT}}. \tag{12}$$

Furthermore, given $\mathbf{y} \in \mathbb{R}^{m \times 1}$ to be the observation vector with $m$ observations from which we can construct additional $N$ perturbed observations $\mathbf{y}_i = \mathbf{y} + \epsilon_{oi}$, and the ensemble of observations $\mathbf{Y}_o \in \mathbb{R}^{m \times N}$ as:

$$\mathbf{Y}_o = (\mathbf{y}_1, \mathbf{y}_2, ..., \mathbf{y}_N) \tag{13}$$

with an associated ensemble of observation perturbations $\boldsymbol{\Gamma} \in \mathbb{R}^{m \times N}$:

$$\boldsymbol{\Gamma} = (\epsilon_{o1}, \epsilon_{o2}, ..., \epsilon_{oN}) \tag{14}$$

from which we can construct the error covariance $\mathbf{R}$:

$$\mathbf{R} = (N-1)^{-1} \boldsymbol{\Gamma} \boldsymbol{\Gamma}^{\mathrm{T}}. \tag{15}$$

The analysis equation then becomes:



$$\mathbf{X}^{\mathrm{a}} = \mathbf{X}^{\mathrm{f}} + \mathbf{K}(\mathbf{Y}_o - \mathbf{H}\mathbf{X}^{\mathrm{f}}); \tag{16}$$

where $\mathbf{H}$ is the (linear) observation operator which relates the forecast model state variables to the measurements. Finally, $\mathbf{K}$ is the Kalman gain:

$$\mathbf{K} = \mathbf{P}^{\mathrm{f}}\mathbf{H}^{\mathrm{T}}(\mathbf{H}\mathbf{P}^{\mathrm{f}}\mathbf{H}^{\mathrm{T}} + \mathbf{R})^{-1}. \tag{17}$$

### 3.3 Covariance inflation

In ensemble methods, such as the EnKF, the analysis step is based on a flow-dependent forecast error covariance $\mathbf{P}^{\mathrm{f}}$, which is a finite size ensemble approximation of the true forecast error covariance (Evensen, 2003). This sampling error causes the matrix $\mathbf{P}^{\mathrm{f}}$ to be usually an underestimation of the actual covariance. Besides sampling error, other sources of incorrect specifications are the model error and the nonlinearities. All these factors may lead to filter divergence, whereby the filter incorrectly trusts a "too small" $\mathbf{P}^{\mathrm{f}}$. Filter divergence is usually handled using covariance inflation (Anderson, 2001; Li et al., 2009; Miyoshi, 2011; Kotsuki et al., 2017; Raanes et al., 2019).

In this study, we use the adaptive covariance inflation method proposed by Desroziers et al. (2006) and based on the following relation:

$$\langle \mathbf{d}\mathbf{d}^{\mathrm{T}} \rangle = \mathbf{H}\mathbf{B}\mathbf{H}^{\mathrm{T}} + \mathbf{R}, \tag{18}$$

where $\mathbf{d} = \mathbf{y} - \mathbf{H}\mathbf{x}^{\mathrm{f}}$ are the innovation vectors, $\mathbf{R}$ is the observation error covariance, $\mathbf{H}\mathbf{B}\mathbf{H}^{\mathrm{T}}$ is the true forecast covariance matrix in observation space, and $\langle \cdot \rangle$ denotes the expected value. This relation holds if $\mathbf{B}$ and $\mathbf{R}$ are correctly known (Desroziers et al., 2006). For an EnKF, we can approximate Eq. (18) using the ensemble-based background covariance matrix $\mathbf{P}^{\mathrm{f}}$ and the inflation factor $\alpha$, according to:

$$\langle \mathbf{d}\mathbf{d}^{\mathrm{T}} \rangle = \alpha \mathbf{H}\mathbf{P}^{\mathrm{f}}\mathbf{H}^{\mathrm{T}} + \mathbf{R}. \tag{19}$$

The inflation factor $\alpha$ is estimated by taking the trace $\mathbf{Tr}(\cdot)$ of Eq. (19):

$$\alpha = \frac{\langle \mathbf{d}\mathbf{d}^{\mathrm{T}} \rangle - \mathbf{Tr}(\mathbf{R})}{\mathbf{Tr}(\mathbf{H}\mathbf{P}^{\mathrm{f}}\mathbf{H}^{\mathrm{T}})}. \tag{20}$$

Since our experiment uses a small sample of observations (i.e., two when the network includes both the ocean and atmosphere and only one when one component is observed), the estimation of $\alpha$ can be plagued by noise, which is detrimental to the correct functioning of the filter (Li et al., 2009). To mitigate the effect of the noise, we used a simple time smoothing function for $\alpha'$ formulated as follows:





$$\alpha'_t = (1-\gamma)\alpha_t + \gamma\alpha_{t-1}. \tag{21}$$

Thus, $\alpha'_t$ is a linear combination of the previous inflation factor $\alpha_{t-1}$ and the one at current time $\alpha_t$; with $\gamma$ being a weighting parameter $0 < \gamma \le 1$, which we tune empirically for the time scale separation $\tau$ of our system.

### 3.4 Assimilation window

We set the length of the ocean (resp. atmosphere) assimilation cycle $\Delta T_o$ (resp. $\Delta T_a$) as one-fifth of the error doubling time:

$$\Delta T = \frac{1}{5}\frac{\ln(2)}{\lambda_1} \tag{22}$$

with $\lambda_1$ being the maximum Lyapunov exponent of the uncoupled L63 system in Eq. (5). This choice reflects observational networks designed or optimized to counteract effectively the error growth (Peña and Kalnay, 2004). Furthermore, and importantly for our goals here, this choice allows us to compare different DA experiments, with varying $S$ and $\tau$, but keeping the

observational constraint at a comparable strength.

As shown in Sect. 2.1, the characteristic time scale of the system is largely modulated by $\tau$ and, to a lesser extent, by $S$ but not by the coupling terms. We therefore decide to determine $\Delta T_{o(a)}$ as a function of $\tau$ only; results are shown in Fig. 9. This relation is exponential, which is expected from Eq. (5) in which $\tau$ is a factor multiplying all variables of the L63 system. Thus, we use a constant $\Delta T_a = 15\,\mathrm{d}t$ for the atmospheric component, as $\tau = 1$ in all experiments (blue dot in Fig. 9). On the other

hand, the assimilation cycle of the ocean $\Delta T_o$ will change with the ocean temporal scale (red line in Fig. 9).

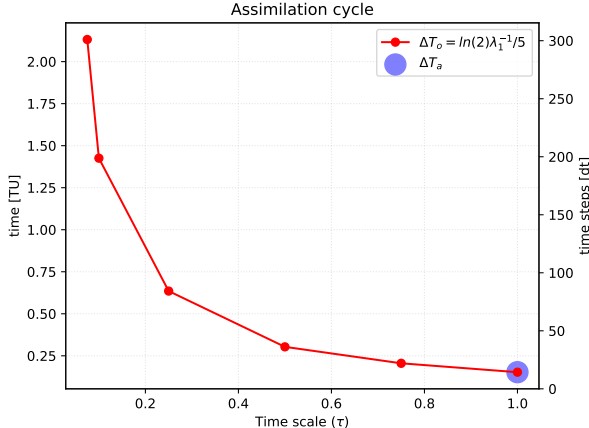

**Figure 9.** Ocean assimilation cycle $\Delta T_o$ (red) as a function of temporal scale $\tau$. The atmosphere assimilation cycle $\Delta T_a$ is marked with the blue circle. Note that the left y-axis shows the assimilation cycle in time units TU, while the right y-axis shows the number of time steps $\mathrm{d}t$.





### 3.5 Evaluation metrics

We assessed the accuracy of each DA approach by computing root-mean-squared error RMSE of the ensemble mean analysis state, averaged over the 30 different experiments, as follows:

$$
\text{RMSE} = \frac{1}{N_e} \sum_{j=1}^{N_e} \left( \frac{1}{N_t - 1} \sum_{i=1}^{N_t} (\overline{\mathbf{x}}_{ij}^{\text{a}} - \mathbf{x}_i^{\text{t}})^2 \right)^{1/2}, \tag{23}
$$

where $\overline{\mathbf{x}}_i^{\text{a}}$ is ensemble mean of analysis state, $\mathbf{x}_i^{\text{t}}$ is the truth at analysis step $i$, and $N_t$ is the integration length. The $j$ indices denote the experiment number performed with a different random seed (here $N_e = 30$, see Sect. 3). In particular, we adopt the $\Delta\text{RMSE}_n$ metric, which uses the WCDA experiment as a benchmark and normalizes the error by that of the FREE run (experiment with no assimilation), calculated as follows:

$$
\Delta\text{RMSE}_n^{\text{SCDA}} = \frac{\text{RMSE}^{\text{SCDA}} - \text{RMSE}^{\text{WCDA}}}{\text{RMSE}^{\text{FREE}}}. \tag{24}
$$

Values of $\Delta\text{RMSE}_n^{\text{SCDA}} < 0$ indicate a reduction of error compared to WCDA, values close to zero indicate no difference, while values $\Delta\text{RMSE}_n^{\text{SCDA}} > 0$ indicate a degradation. We present the results as the mean of $\Delta\text{RMSE}_n^{\text{SCDA}}$ over each component. Hence, we present the average error reduction $\Delta\overline{\text{RMSE}}_n^{\text{SCDA}}$ in the full atmosphere and ocean.

## 4 Results

We compare WCDA and SCDA in a set of numerical experiments grouped based on the observational network. Specifically, we
shall have experiments with observations in both (i) the atmosphere and ocean (named $FULL$ hereafter), (ii) the atmosphere ($ATM$), and (iii) the ocean ($OCN$) — see Sect. 3 for a detailed description of the experimental design.

### 4.1 Joint atmospheric and ocean observations network (FULL)

A comparison of WCDA with SCDA with observations on ocean and atmosphere components is shown in Fig. 10. It is important to note that overall, the differences between the SCDA and WCDA are very small (about 0.1 % of climatological
error); i.e., both systems perform nearly equally well. It was already reported in Sandery et al. (2020) that the SCDA benefit over WCDA reduces when both components are well observed. While confirming that finding, our results further demonstrate that WCDA performs slightly better than SCDA in most spatio-temporal configurations.

In the atmosphere (Fig. 10a), SCDA yields slight degradation as the temporal scale separation decreases and most when the scale separation between the two components is large (small $S$). Both components are highly chaotic when $\tau$ is 1 (Fig. 7). When
$S$ is small, most of the energy is found in the chaotic ocean component (Fig. 3a). There is little gain during the assimilation as the cross-covariance of the system is relatively small (Fig. 4), and the performance is highly sensitive to spurious covariance





in the system (sampling error). This result also aligns with the linear error analysis carried out in Tondeur et al. (2020) that suggested that when the temporal scale separation is not large, WCDA is preferable.

In the ocean (Fig. 10b), the dependence of the error on the temporal separation is the opposite of that seen in the atmosphere.

It increases as the time scale separation gets larger, i.e., as the "stable" ocean gets more sensitive to the chaotic atmosphere. When the energy of the system is dominated by the ocean (small $S$), performance is slightly degraded, but when the energy is distributed or prominent in the atmosphere component ($S > 1$), and the cross-covariance is maximum, the SCDA improves over WCDA.

SCDA has little advantage over WCDA when both components have good observation coverage. Furthermore, the system

becomes vulnerable to the approximation inherent in the DA (sampling error, linear analysis update), which can lead to slight degradation. We speculate that if we were increasing ensemble size and softening the linear analysis update (e.g., with iterative approaches), this degradation would be gone, and the SCDA would outperform WCDA, but discrepancies would remain marginal.

Note also that when the ocean and atmosphere assimilate data every $20$ dt, SCDA improves over WCDA for all $(S, \tau)$ con-

figurations, especially for the ocean, but the difference is again very small (not shown). This result agrees with the experiments of (Penny et al., 2019), which shows that SCDA provides better analyses than WCDA in a fully observed system, with the same assimilation cycle on both components.

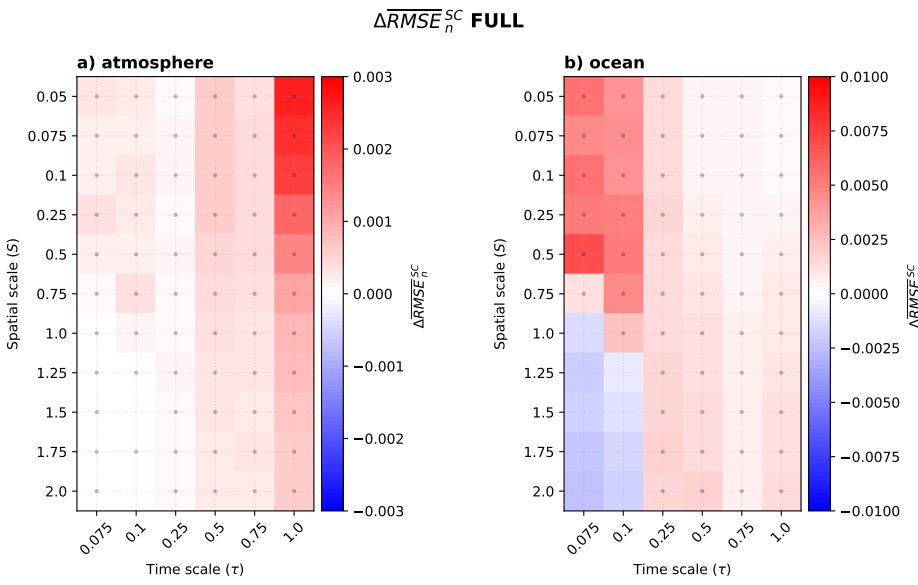

**Figure 10.** FULL experiment $\Delta \overline{\mathrm{RMSE}}_n^{SC}$ for a) atmosphere and b) ocean. The colour red (blue) indicates that the SC error is larger (smaller) than that of the WC experiments. Dotted area indicates $\Delta \overline{\mathrm{RMSE}}_n^{SC} \geq 0$, meaning that SCDA degrades over WCDA. Note that the colourbar for both components has different limits.





## 4.2 Atmospheric observation network (ATM)

When we observe only the atmosphere (Fig. 11), SCDA shows improvement over WCDA in nearly all spatio-temporal scale
experiments. The pattern of improvements in the atmosphere and ocean components is similar; however, the ocean shows
comparatively a larger improvement.

The benefit is largest when $\tau$ gets small as the ocean gets less chaotic and well constrained by the atmospheric state. The
region for $\tau < 0.25$ shows a very strong sensitivity to $S$. Improvement of SCDA is largest when the ocean has a large scale
($S < 0.75$), and holds most of the system's energy. The well-constrained atmosphere and atmospheric data (even with moderate
cross-covariance) can constrain the predictable ocean. In the region where the ocean has a smaller scale ($S \geq 0.75$), SCDA
has no impact, unlike in the *FULL* experiment. This result can be explained as the *atmosphere $\rightarrow$ ocean* error propagation is
smaller (Fig. 8); thus, the atmospheric data has no impact over the ocean. Indeed, the lack of ocean data, implies that the ocean
cannot influence the atmosphere at analysis time. Furthermore, the ocean's energy is too small to impact the coupling during
the model integration step.
Thus, we can conclude that when only the atmospheric component is observed, it is highly beneficial to use SCDA, except
when the time scale separation is very large and the energy in the non-observed component is small.

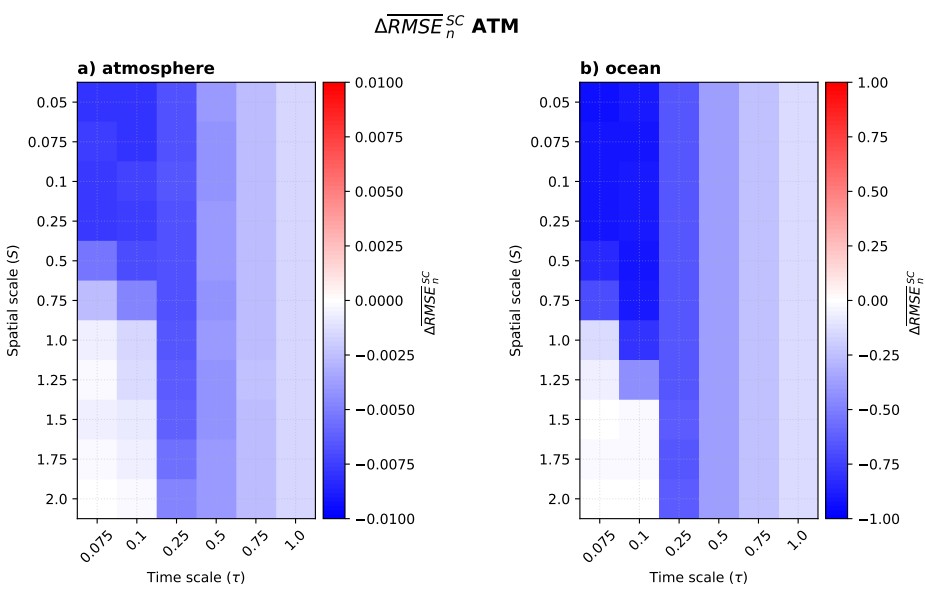

**Figure 11.** As for Fig. 10 but with only the atmospheric state observed — ATM experiment.



### 4.3 Ocean observations network (OCN)

When observing only the ocean, the skill pattern of the SCDA, Fig. 12, is overall similar to what is seen in the case of only the atmosphere being observed. Nevertheless, we see now that the impact (positive or negative) of SCDA is larger in the atmosphere and almost negligible in the ocean ($\sim 1$ % of the climatological error).

Strongly coupled DA degrades over WCDA when the ocean is fast and has a large amplitude ($S \geq 0.25, \tau \geq 0.5$). Both components are chaotic, but the ocean holds most of the system's energy. As the cross-covariance is minimal, and the *ocean $\rightarrow$ atmosphere* error propagation is small, the usefulness of the cross-update from the ocean towards the chaotic atmosphere is limited and sensitive to the linear update and sampling error. On the contrary, when the ocean's energy decreases ($S < 0.25, \tau \geq 0.5$), the ocean state benefits from the improved atmosphere's initial condition during the model integration. There, the cross-covariance is high with an increased *ocean $\rightarrow$ atmosphere* error propagation so that the atmosphere can benefit from ocean observations. These results resemble those obtained in (Tang et al., 2021), which uses SCDA in a real framework to update the atmosphere with ocean observations, improving the ocean-atmosphere tropical interface.

It is somewhat surprising that little improvement in the atmosphere is found when it interacts with a slow and small ocean ($S \geq 0.25, \tau < 0.5$). In this region, the error propagation is largely in the atmosphere, and the cross-covariance maximum and the sensitivity to error growth are largely dominated by the *ocean $\rightarrow$ atmosphere*. We conjecture that the ocean assimilation cycle is so large that the cross-update cannot properly constrain the atmospheric state, in agreement with (Tondeur et al., 2020).

In a way, the conclusions of the ocean observation network are quite analogous to those found with the atmospheric observation network (being symmetrical w.r.t the spatial scale). We could anticipate that when the ocean time scale gets as fast as the atmosphere, the benefit will further increase until a certain threshold.

## 5 Concluding remarks

### 5.1 Summary and main findings

This study investigates how spatio-temporal scale separation in coupled atmosphere-ocean dynamics and the availability of observational data on either or both of the components influence the skill of different approaches to coupled data assimilation. In particular, we analyze the so-called weakly and strongly coupled data assimilation (Penny and Hamill, 2017). In the former, the observations in one of the model components are used to infer the state of that component only during the assimilation step. In the SCDA, the observations are all used to infer the full coupled model, no matter where they are taken.

We focus on ensemble-based data assimilation using the well-established EnKF (Evensen, 2003) and use a prototypical low-dimensional coupled system obtained by coupling together two Lorenz-63 models with different parameters (Peña and Kalnay, 2004). The model configuration allowed us to explicitly modify the parameters affecting the spatio-temporal scales, let alone that the model's low computational cost made it possible to obtain statistically robust results.

The main conclusions are the following:





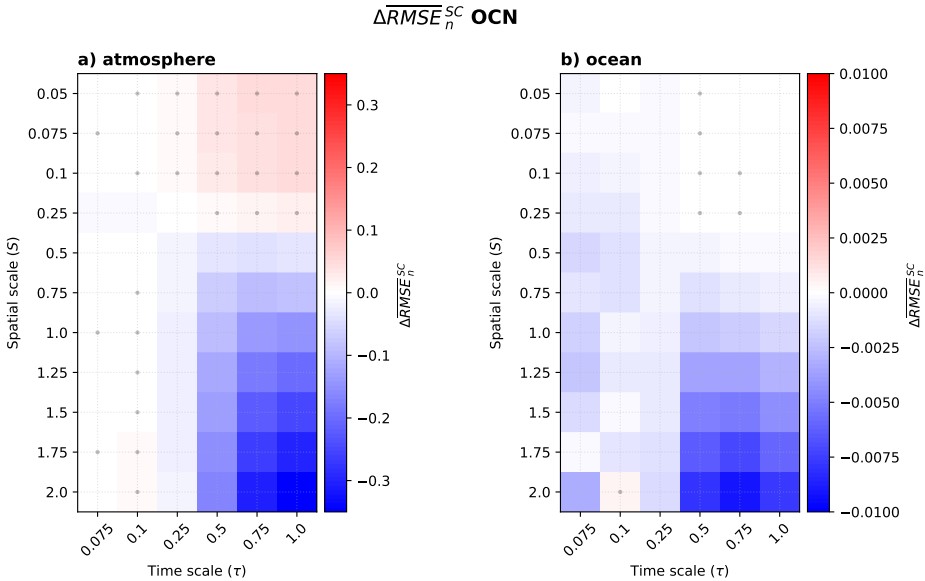

**Figure 12.** As for Fig. 10 but with only the ocean state observed — OCN experiment.

1. In a well-observed system, the potential for improvements over WCDA is very limited as observations from both components constrain the system nearly optimally. With our implementation (EnKF), we even find that SCDA degrades the system's performance very slightly due to approximation in the DA method — linear analysis update, sampling error.

2. SCDA improves over WCDA when only one component is observed, and improvements are largest in the non-observed component. The benefit is larger when the observed component has a smaller spatial scale (hence less energy in our idealized experimental setup). As such, the slow-but-large variability modes of the ocean can be improved from atmospheric observations, which, in turn, improves dynamically during the forecast step. Similar conclusions have also been drawn in other idealized studies such as Han et al. (2013), Sluka et al. (2016), and Tondeur et al. (2020), in which the assimilation of atmospheric observations improves ocean reanalysis. Conversely, using the relatively fast-ocean observations, it is possible to constrain fast-large atmosphere modes. In this regard, already O'kane et al. (2019) and Sandery et al. (2020) in experiment $A^O O^A$, indicated that the assimilation of ocean observations improves the initialization of a tropical-atmosphere component.

## 5.2 Discussion

Our study uses a low-order complexity model to explore isolated cases of spatio-temporal scale separations with different observation networks. Despite its simplicity, our study confirms previous studies' findings and provides a complete picture of configurations where SCDA is expected to yield improvement over WCDA. However, we acknowledge the simplifications of





our experiment design and discuss their expected impact on our conclusions: effects of the coupling strength, the superposition

of spatio-temporal scales, the choice of data assimilation method, the choice of the observation network and model biases.

Our experiment assumes a weak coupling and studies coupled processes in isolation, while the coupling strength varies from process to process and often influences each other. The influence of the coupling strength for comparing SCDA and WCDA was studied in Tondeur et al. (2020), where it was shown that a stronger coupling results in a more stable system and higher cross-covariances among the components. As such, we can anticipate that as the coupling gets weaker, the observed benefit

of SCDA over WCDA will fade out and strengthen with a stronger coupling. We do not think combining several processes is an issue, as standard DA methods (particularly ensemble methods) are designed for that. However, it is expected that a larger ensemble size is required.

All experiments in our study were carried out with the EnKF, which has the advantage of providing flow-dependent error covariance, a property that is important for SCDA (Penny et al., 2019). However, the linear analysis update in the EnKF

is suboptimal for strong non-linear dynamics (Sakov et al., 2012; Yang et al., 2012), which can also occur with a too-long assimilation cycle. In our study, we encountered these situations when disparate spatial scales and strongly chaotic components led to a degradation of SCDA. We expect that methods that soften the linear analysis update approximation, such as the iterative Ensemble Kalman Filer (iEnKF, Bocquet and Sakov, 2012, 2014; Evensen et al., 2024) or outer-loop 4D-Var (Laloyaux et al., 2016, e.g., CERA-like method) would improve the performance of SCDA. The iEnKF has been tested in a low-complexity

coupled model with different spatial scales by Evensen et al. (2024), showing a good performance. It would be interesting to test whether the iEnKF could remove the degradation of the SCDA over the WCDA with our system. In situations when the iterative approach is unable to mitigate the approximation from the linear update, one could consider using the lagged cross-correlations between the system's components as proposed by Lu et al. (2015a, LACC method). It should be acknowledged that the LACC method only allows for a way stream of information (from the fast to the slow component) and is challenging

to use with the superposition of processes with different spatio-temporal scales.

Another limitation of the EnKF is sampling error, as large ensemble sizes are needed to accurately sample the variance of the system (Han et al., 2013; Quinn et al., 2020). We did not investigate this issue in our study as 20 members can already be considered a large ensemble size given the size of our dynamical system (Bocquet and Carrassi, 2017). With a realistic system, the sampling error is comparatively much larger. Based on our dynamic analysis, we can infer that the limited ensemble

size becomes a more restrictive factor for the successful implementation of SCDA as the time scale separation increases (Fig. 6). Different approaches have been proposed to address sampling error. Some methods consider the system's dynamic characteristics, such as Quinn et al. (2020), which uses the attractor dimension to estimate the rank of the cross-covariance needed for SCDA. Ad-hoc solutions such as vertical localization, as discussed by Stanley et al. (2024), or the correlation-cutoff method, as in Yoshida and Kalnay (2018), and hybrid covariance (Barthelemy et al., 2024) are also options to address the same

issue.

Our study highlighted that the observational network is a significant factor in when SCDA outperforms WCDA. Here we only asses configurations where observations are only in the atmosphere, the ocean, or in both components. The situation is not as distinct in a real framework but still shows a strong imbalance between the observation network of the different



components. Historically, ocean observations have been scarce compared to the atmospheric network (Laloyaux et al., 2018). We can thus anticipate that the largest benefit of SCDA is expected in the ocean component, meaning that the slow ocean modes of variability can benefit from the high-frequency atmospheric variability, as shown in Sandery et al. (2020). However, we acknowledge that there has been recent rapid progress in the ocean observation network. For example, the SWOT altimetry (Morrow et al., 2019) may allow to constrain ocean fronts at a much finer scale than is currently possible, providing an exciting perspective on SCDA having the capability to enhance numerical weather prediction based on ocean observations.

Finally, given our perfect-model assumption, we have not addressed one critical aspect: how model bias can hinder SCDA's potential. Earth System Models have large biases (Palmer and Stevens, 2019; Richter et al., 2014; Tian and Dong, 2020), and coupled processes are often only partially represented. Therefore, as models increase their resolution and the availability of ocean observations changes and better resolve those coupled processes (Hewitt et al., 2017; Kim et al., 2023), ocean observations can provide more useful information about the atmosphere's surface processes, making SCDA a powerful tool for initializing the coupled system and making skilful predictions.

*Code and data availability.* The code used in this study is available from the corresponding author by request.

*Author contributions.* All the authors authors contributed to the study's conception and design. LG-O implemented the statistical and dynamic study in Python. LG-O produced the Data Assimilation codes and evaluation. All authors participated in the study, discussing results, writing, and approving this manuscript.

*Competing interests.* At least one of the (co)-authors is a member of the editorial board of Nonlinear Processes in Geophysics.

*Acknowledgements.* This study was partly funded by the Trond Mohn Foundation, under project number BFS2018TMT01, the NFR INES ((INES; 270061), and Climate Futures (309562). Finally, we acknowledge the Nansen Center's foundational institutional funding, made possible by the Research Council of Norway grant #342624. AC acknowledges the support of the project SASIP, which was funded by Schmidt Sciences (Grant number 353). Schmidt Sciences is a philanthropic initiative that seeks to improve societal outcomes by developing emerging science and technologies. We would like to thank Patrick N. Raanes and Sebastien Barthelemy for their invaluable help and discussions while developing and implementing the code.



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
