# Peer review of "Exploring the influence of spatio-temporal scale differences in Coupled Data Assimilation"

_EGUsphere, 2024_

## Author Comment (AC1)

**Reply to the reviewer's comments**

Lilian Garcia-Oliva, Alberto Carrassi, and François Counillon September 2024

**Manuscript No.: EGUSPHERE-2024-1843**

Title: "Exploring the influence of spatio-temporal scale differences in Coupled Data Assimilation"

Thank you for your constructive and helpful comments. We provide our responses in blue, while for the proposed modifications to the manuscript we use **bold** text. We also provide the line numbers from the new-corrected manuscript.

**Reviewer #1:**

This manuscript explores the optimal strategy for initializing coupled climate prediction systems by comparing between strongly and weekly coupled data assimilation. Through a series of experiments, the authors have reach to the two conclusions described in the last section. While the conclusion is reasonable to me, I am not fully convinced of the significance of the manuscript. The conclusions presented in Chapter 5 were largely consistent with previous studies. This also indicates that the findings obtained in this study are quite limited. Consequently, I cannot be very positive to the paper because this study seems to be a simple extension of previous studies. I have some suggestions for improving the manuscript, as described below.

**R:** We thank the Reviewer for their comments and criticism. We recognize now that we did not convey adequately the motivations and the novelties in the original version of our manuscript. We hope to be able to rectify this in our revised version. Specific answers to your queries are provided below.

In our opinion, existing literature on the added value of weakly over strongly coupled DA (WCDA and SCDA, respectively) has often been discordant, with some concluding about improvements and others showing a degradation depending on the configuration. Several factors can influence the conclusions: e.g. spatio-temporal scale separation, model error, sampling error, ad-hoc fixes (e.g. localization).

Our main research objective is to unveil the possible connections between the underlying dynamical properties of the forecasting model and the performance of coupled DA. For this reason, our study intentionally leverages a simple, yet nonlinear and chaotic, system to thoroughly assess the performance of WCDA and SCDA in a wide range of spatio-temporal scale separation that are observed in the real climate system. The identical twin experiment setup allows us to eliminate model bias, the low computational cost made possible the use of a large ensemble size to avoid or mitigate sampling error. Furthermore, having a full control on the model's dynamical features we were able to compare a wide range of parameter configurations while keeping the average rate of error growths within the same level. The latter fact has been pivotal to accomplish a comparison among various model configurations where the interest was in understanding the impact of varying the spatio-temporal scale separation

across model compartments rather than on the effect of changing observation interval (which was always set to a given, fixed, multiple of the error doubling time).

Another key consequence of our idealized experimental setup is that we did not need localization, otherwise necessary in higher dimensions. By modifying the error covariances, localization breaks dynamical consistency (in return it provides statistical appropriateness) thus rendering extremely intricate to disentangle the role of spatio-temporal scale separation in shaping the error covariances.

Previous literature on the topic has addressed CDA in specific settings. To the best of our knowledge, our work is the only study that analyses the issue with both spatial and temporal scale separation and observation networks. For instance:

Tondeur et al. (2020) analyses the impact of the temporal scale separation on each CDA approach. This paper explores the observation network and assimilation frequency using a fixed temporal scale separation. On the other hand, Evensen et al. (2024) analyses spatial scale separation. Furthermore, the study by Miwa and Sawada (2024) explores the CDA approaches using several model configurations, such as coupling strength and three different temporal scales, using a fixed spatial scale. In this study, they managed to link the relationship between coupling strength and the chaoticity of the system. In this context, our study's originality roots in the explicit combination of several spatio-temporal scale separations, observational networks, and the dynamics of the underlying system to better understand CDA performance.

(1) Usage of more complex model(s): One of the main discussions in coupled data assimilation is how to differentiate real and erroneous error covariance. Therefore, exploring a better localization strategy is essential for coupled data assimilations. However, the present manuscript uses a very simple top model, which is unsuitable for investigations on localization. It is also important to investigate optimal observation frequency (=data assimilation) of the fast and slow-mode models for the coupled data assimilation.

**R:** We agree that localization is essential to the success of CDA in a realistic, high-dimensional, framework. Nevertheless, the main objective of our manuscript is to link the spatio-temporal scale separations to the performance of CDA. With that scope in mind, we see it as a strength that the system does not require localization. We could thus analyse the effect on CDA of the different degree of instability and spatio-temporal scale separation in the dynamical model. Localization would hide the effects we aim at studying.

We also agree that the assimilation frequency has a large influence on the results. Precisely in the light of this, we have designed our comparison of CDA at different spatiotemporal scales by changing the observation frequency such that the interval between successive observations was fixed to 1/5 of the error doubling time (proportional to the first Lyapunov exponent). As such we ensure that all experiments are compared at similar error level and that the differences only relate to the scale separation.

(2) To investigate various coupled data assimilation strategies: Kurosawa et al. (2023; NPG) investigated various options of coupled data assimilation, as indicated in Figure 2. I would suggest investigating such options together with the sensitivity investigations on observation frequency, ensemble size and localization.

**R:** We thank the Reviewer for pointing out a paper that we did not consider in our original manuscript, which we are now including as one of our references. In the study by Kurosawa et al. (2023) the SCDA and WCDA experiments are evaluated in a system using atmosphere and land observations. Since in our study we use an atmosphere-ocean system, we consider atmosphere and ocean (instead of land) observations. Most of the assimilation strategies proposed by Kurosawa et al. (2023) were already considered in our study; however, there were two (2) configurations that we did not consider. Using Kurosawa's notation for the experiments, the 'missing' configurations are  $A_{A\times}L_{AL}$ , and  $A_{AL}L_{\times L}$  (both SCDA); where A and L indicate Atmosphere and Land, respectively. These configurations correspond to cases in which both components are observed but only one cross-update is performed. For example, the configuration  $A_{A\times}L_{AL}$  considers a well observed system (observations for atmosphere and land) thus, performing atmosphere DA and land DA with the additional cross-update from the atmosphere to land. The same applies for the other experiment.

In the context of our experiment, these cases correspond to our SCDA-FULL experiments, with only one cross-update, from the atmosphere or from the ocean. We considered the same experimental settings of the cases shown in the manuscript. The new cases are:

- FULL-A: Atmosphere DA, ocean DA and atmosphere  $\rightarrow$  ocean cross-update.
- FULL-O: Atmosphere DA, ocean DA and ocean  $\rightarrow$  atmosphere cross-update.

We decided to analyse these cases, comparing them with our WCDA-FULL experiment. This comparison will reveal the impact of only the cross-update. We show our results in Figures 1 and 2, with the metric  $\Delta \overline{\text{RMSE}}_n^{\text{SCDA}}$  indicating the averaged difference in RMSE between WCDA and SCDA.  $\Delta \overline{\text{RMSE}}_n^{\text{SCDA}} > 0$  indicates that SCDA has larger error than WCDA.

These results (Figures 1 and 2) show that in a fully observed system, the additional cross-update from SCDA causes a suboptimal update, therefore the use of WCDA is better than SCDA. These results reinforce the idea that the estimated cross covariance, used for SCDA, suffers from the sampling error, and that the linear analysis update of the EnKF is suboptimal for the cross-update (see our answer to Reviewer 2 on this point). The sensitivity is higher as the temporal scale separation increases, especially on the observed component.

*Figure 1:* FULL-A experiment metric  $\Delta \overline{\text{RMSE}}_n^{\text{SCDA}}$  for a) atmosphere and b) ocean. Red indicates that the SCDA error is larger than the WCDA. Note that the colourmap has different limits, all positive.

*Figure 2:* Figure 1FULL-O experiment metric  $\Delta \overline{\text{RMSE}}_n^{\text{SCDA}}$  for a) atmosphere and b) ocean. Red indicates that the SCDA error is larger than the WCDA. Note that the colourmap has different limits, all positive.

**Reviewer #2:**

The authors tried to clarify when the strongly coupled data assimilation (SCDA) is preferrable to weakly coupled data assimilation (WCDA) by a two-components coupled Lorenz-63 system (one component representing the atmosphere, fast component; the other one representing the ocean, slow component), with changing the parameters of observation networks (FULL, ATM, OCN), spatial scales (S, 0.05 - 2.0), and temporal scales (\tau, 0.075 - 1.0). The data assimilation implemented in this study is EnKF with perturbed observations and adaptive inflation. In WCDA, the assimilation is applied to the individual components separately by

using the observations available for that component. In SCDA, the observations from one component impact the other components directly during the assimilation.

The manuscript described the stability analysis of the coupled Lorenz-63 model, which is a simpler version of Tondeur et al. (2020). The results were discussed with respect to the observation networks: (1) in a well-observed system, SCDA degrades the system's performance slightly compared to WCDA; (2) SCDA improves over WCDA when only one component is observed. Similar conclusions have been reported by previous studies. I think the interpretation from the instability analysis on why the SCDA shows different responses from WCDA would be interesting to the readers, whereas the results and conclusions of the paper are not consistent. My concerns are listed as followings.

**R:** Thank you for your constructive comments and the careful reading of our manuscript. The Reviewer has also well understood our main goal of connecting the CDA performance to the underlying instability properties of the dynamic model. Prompted by her/his criticism, in our revised version of the manuscript we have attempted to improve our discussion and strengthen the interpretation of such connections. We have also tried to clarify the potential inconsistencies as highlighted by the Reviewer.

**Major comments:**

1. Lines 5 - 7. (a) In full observations, "SCDA and WCDA yield similar performances". Does this mean that the spatial scale (S) and temporal scale (\tau) have very few influences on the coupled data assimilation? (b) If the SCDA performs marginally worse than WCDA could be explained by the approximation in the EnKF – linear analysis update and sampling error, I will encourage the authors to explicitly describe how the linear analysis update and sampling error in the SCDA differs from those in the WCDA.

**R:** We are grateful to the Reviewer for pointing to these potential inconsistencies. Accordingly, we have revised the text to make these statements more precise and discuss them clearly in our results and discussion section. We respond to your comments:

- a) In our original abstract (Lines 5-7) we wrote that in the FULL experiment "SCDA and WCDA yield similar performances" because the difference between each method is around 0.1% and 1% of the climatological (FREE) RMSE for atmosphere and ocean, respectively. Therefore, we consider that both methods perform similarly. Nevertheless, the performance, albeit generally similar, clearly depends on the spatio-temporal scales (S and \tau values); we discussed this in greater detail in Sect. 4.1.
- b) SCDA provides a slightly worse update than WCDA in the FULL coverage observation network. We speculate that this difference is due to the approximations done with the EnKF; the linear analysis update and sampling error. In SCDA, the presence of sampling error is larger relative to the size of the state vector -- in SCDA, the state vector is 6 whereas it is 3 in WCDA for the update of the individual components. Also, the cross-component covariances are small (so more prone to sampling error) and becoming more non-linear as the scale separation increase. A linear analysis can lead to a degradation and non-linear iterative approach should be more suitable.

In the revised version of the manuscript, we modify our abstract and main conclusion to account for the influence of a) the spatio-temporal scales and b) the impact of the approximation inherent to the EnKF; the revised text now reads:

The abstract:

Lines 5 - 10: "In the fully observed scenario, SCDA and WCDA yield similar performances. However, some little differences are present, and we conjecture these are due to the SCDA being more sensitive to the approximations at the basis of the EnKF present in the cross-update – linear analysis update and sampling error. This sensitivity increases as the temporal scale separation increases, especially on the slow-large scale component."

We also updated our first conclusion from our Summary and main findings (Sect. 5.1) as:

Lines 363 – 369: "In a well-observed system, the potential for improvements over WCDA is very limited as observations from both components constrain the system nearly optimally already. We even find that sometimes SCDA degrades the system's performance. This is possibly due to the approximation in the DA method – linear analysis update and sampling error. The state vector during the assimilation with SCDA is 6 whereas it is 3 with WCDA for the update of the individual components. Consequently, the sampling error is larger relative to the dimension of the state vector with SCDA than with WCDA. Furthermore, the cross-components covariances are often weaker and their non-linearity growth as the scale separation increases. The linear approximation during the analysis with the EnKF can yield a degradation. When the time scale separation (and to a lesser extent the spatial scale separation) is large, a nonlinear update (e.g. Evensen 2024) may be better suited."

2. Lines 7 - 8. When observations are only in one of the components, "SCDA systematically outperforms WCDA" is contradictory with the discussions in Chapter 4.3 and Figure 12 (a), where the dotted area means SCDA degrades over WCDA (Figure 10).

**R**: We thank again the Reviewer for pointing to a possible overstatement. From the figures that you mention it is obvious that, for the atmosphere in our OCN experiment (Fig. 12a), SCDA's improvement over WCDA has a clear dependence on the spatio-temporal scale separation. Thus, we modified our conclusion as:

Lines 10-12: "When observations are only in one of the components, the spatio-temporal scale separation determines SCDA's performance. In this scenario, the largest improvements are found when the observed component has a smaller spatial scale..."

3. Lines 8 - 10. "The spatio-temporal scale separation determines SCDA's performance in this scenario, and the largest improvements are found when the observed component has a smaller spatial scale." (a) The first part of this sentence says that the spatio (S) -temporal (\tau) scale affects the SCDA's performance, whereas the second part says that only the spatial scale (S) affects the performance, which is not consistent. (b) In Figure 12, when only the ocean was

observed, the large improvements were found when the spatial scale is larger, which is contradictory with the sentence in the abstract.

R:

a) We clarified this apparent contradiction. Now it reads:

Lines 11 - 13: "When observations are only in one of the components, the spatiotemporal scale separation influences SCDA's performance. In this scenario, the largest improvements are found when the observed component has a smaller spatial scale. The fast-to-slow update has a larger benefit with a larger temporal scale separation. Meanwhile, with the slow-to-fast update, the improvement is limited to instances when the temporal scale separation is less than one-half."

b) We understand the confusion of the reviewer. The actual spatial scale of the ocean is inversely proportional to the parameter S (Sect. 2.1), which we estimated using the 'Energy ratio' between the two components (Fig. 3a). In Fig. 12a the largest improvements in the unobserved atmosphere occur when  $S \ge 1$  and it is maximum at S = 2, \tau = 1. This corresponds to the configuration where **the observed ocean has a smaller spatial scale** (relative to the atmosphere) and similar time scale. It is thus, in agreement with the abstract. To clarify this, we modified our manuscript in Sect. 2.1, to explicitly indicate that the ocean's spatial scale is inversely proportional to S. The manuscript now reads:

Line 102 - 114: "We use energy E to estimate each component's spatial scale. The energy E of the two components ... The relative energy content of each component (Fig. 3a) shows that the energy of the ocean  $(E_o)$ , and hence the spatial scale separation, is mostly inversely proportional to S and that the temporal scale has only a little influence on it. Therefore, the ocean's spatial scale increases as S > I."

4. Lines 10. "This suggests that SCDA of fast atmospheric observations can potentially improve the large-slow ocean component." This sentence is contradictory with the discussion in the Chapter 4.2, Lines 326 - 327: "This result can be explained as the atmosphere  $\rightarrow$  ocean error propagation is smaller (Fig. 8); thus, the atmospheric data has no impact over the ocean."

**R:** The discussion in Sect. 4.2 on the lines 326-327 refers to the regions where the spatiotemporal scale parameters are  $S \ge 0.75$  and  $\tau

---

## Author Response (AR2)

**Reply to Reviewer #3's comments**

**Title:** "Exploring the influence of spatio-temporal scale differences in Coupled Data Assimilation"

Manuscript No.: EGUSPHERE-2024-1843

Written by: Lilian Garcia-Oliva, Alberto Carrassi, and François Counillon

"Exploring the influence of spatio-temporal scale differences in Coupled Data Assimilation" by Garcia-Oliva and co-Authors is a remarkably well-written and well reasoned article on the topical issue of what is the best level of coupling in Earth System DA, specifically Atmosphere-Ocean coupling. The model used in the simulation is very simple (Two components Lorenz 63), however the examination of its behaviour from a dynamical system perspective is thorough and lucid. The examination of how the system behaves in an EnKF DA framework is also informative and well reasoned.

Thank you very much for your encouraging comments and insightful suggestions for improving our manuscript. We provide our responses in blue, while for the proposed modifications to the manuscript, we use **bold** text. We also provide the line numbers from the new, corrected manuscript.

On the other hand, I believe the paper could be significantly strengthened in two directions:

1) One is to add to the discussion a new baseline system where DA is performed independently in the Atmosphere and Ocean and also the forecast step is run an uncoupled mode. In other words, compare WCDA with an uncoupled DA system. This is useful because WCDA of the Atmosphere-Ocean system is currently run in major NWP Centres and comparing results from the toy model used in this work with results in operational coupled system would validate methodology and applicability of the results presented in this paper about the potential impact of SCDA vs WCDA. For example, the experience in the operational community about the forecast impact of WCDA with respect to an uncoupled baseline has been that forecast impacts are generally small, transient and localised near the interface (e.g., Browne, P. A., et al., 2019. Weakly Coupled Ocean–Atmosphere Assimilation in the **ECMWF NWP** System. Remote Sensing, 11(3), 234. https://doi.org/10.3390/rs11030234);

**R:** We agree with the reviewer that the UCDA vs. WCDA is an important discussion, especially in the context where most operational centres are moving toward a WCDA initialisation. Prompted by the Reviewer's comment, we include now some UCDA experiments and discussion in the revised manuscript. The modifications shall be:

**Abstract**

In 16 − 19: "... case, the cross-updates may become too sensitive to data assimilation approximations. We further validated that WCDA systematically outperforms uncoupled data assimilation (UCDA) in both components, legitimizing the transition toward WCDA."

**Introduction**

In 39 – 44: "... information on the error terms (Carrassi et al., 2018). Traditionally, NWP and S2D predictions are initialized using uncoupled DA (UCDA). UCDA consists in the realization of independent data assimilation cycles on each of the relevant components of a coupled model (Meehl et al., 2021). However, when applying UCDA to coupled models, it often results in imbalances between the ocean and atmosphere states, which causes initialization shock and reduces prediction skill (Balmaseda et al., 2009; Zhang et al., 2020). To alleviate such limitations, coupled DA (CDA) is produced with..."

**In 63 – 66:** "Our study's main motivation is to explore potential connections between the coupled model's dynamical properties and the **performance of uncoupled and coupled DA methods**, and how the latter interplay with the spatio-temporal scale separation among the model's components."

In 67 – 71: "We use a low-order coupled system and extensively compare the different approaches for a wide range of temporal and spatial scales and observation configurations. This can help us to verify when CDA - particularly WCDA - is expected to outperform UCDA, and further anticipate when SCDA is expected to outperform WCDA in an operational configuration, thus legitimizing the allocation of resources to migrate from UCDA to WCDA, or all the way to SCDA."

**Data Assimilation Experiments**

In 211 – 215: "We conduct a set of coupled and uncoupled DA experiments using the coupled L63 in Eq. (1), and the uncoupled L63-like system in Eq. (5), respectively, for different values of the parameters S and  $\tau$  to reflect spatio-temporal separations between the two components of the system. We used the stochastic Ensemble Kalman Filter (EnKF Evensen, 2003) and compared weakly and strongly coupled data assimilation (WCDA and SCDA, respectively). The additional experiment using UCDA is contrasted against WCDA."

**Evaluation metrics**

In 303 – 309: "For the comparison between UCDA and WCDA, we use the metric \$\Delta \ text{RMSE}^{\text{UC}}\_n\$, defined similarly to Eq. (24), but now evaluating the UCDA experiment, thus:

$$\Delta \text{RMSE}_{n}^{\text{UC}} = \frac{\text{RMSE}^{\text{WCDA}} - \text{RMSE}^{\text{UCDA}}}{\text{RMSE}^{\text{FREE}}}$$

In this comparison, the RMSE for UCDA is computed using the truth and FREE run, both calculated using the coupled L63 system in Eq. 1. The metric \$\Delta \text{RMSE}^{\\}

text{UC}}\_n\$ assesses the capability of UCDA to reconstruct the variability of one of the components of a coupled system, using its uncoupled version."

**Results**

In 312 - 315: "... We also show the comparison between UCDA and WCDA under the FULL observation network. This is, we compare both methodologies under a well-observed system — i.e., observing the (y, Y) variables in each component. Thus, we have UCDA-A and UCDA-O, uncoupled DA in the atmosphere and the ocean, respectively."

ln 316 – 329:

**"4.1 Uncoupled versus weakly coupled data assimilation**

Figure 10 shows the error of UCDA compared to that of WCDA. In general, UCDA gives larger errors in both components, indicating that using the coupled model for forecasting is useful for propagating information across model compartments and further decreasing the error. The error of UCDA is different on each component, with the ocean presenting the larger difference between UCDA and WCDA.

We can see that in UCDA-A (UCDA in the uncoupled atmosphere, Fig. 10a), the error has approximately the same magnitude across all the spatio-temporal scale separations. On the other hand, the error in UCDA-O (UCDA in the uncoupled ocean, Fig. 10b) shows a clear pattern of increasing error toward the small-slow modes of variability. Since the same pattern is observed when comparing UCDA-O with a partially observed WCDA — i.e. when observing atmosphere or ocean only — (not shown), we can conclude that the coupling is key to further decrease the error growth, via the system's dynamics. This pattern in the ocean becomes evident due to the dynamic characteristics of the system. The area where UCDA-O performs the poorest is a region where the cross-component correlation is largest (Fig. 4), and the dominant error propagation is ocean → atmosphere (Fig. 8); therefore, the interaction between both components is vital for efficient error constraint, especially in the small-slow modes of ocean variability. This shows that a coupled analysis provides better assimilation"

**Concluding remarks**

**In 394 - 396:** "... In particular, we analyze the **so-called uncoupled, weakly and strongly** coupled data assimilation (Penny and Hamill, 2017). **In uncoupled data assimilation, observations are assimilated using an uncoupled system.** In the WCDA..."

In 405 - 408: "The coupling between the system's components is vital for error constraint, and its consideration that is possible in the coupled data assimilation framework provides an effective method for decreasing initial error compared to UCDA. In particular our findings indicate that the ocean is important for atmospheric improvement, as noted by Browne et al. (2019), and the atmosphere-ocean interactions become increasingly important for constraining the ocean's slow variability."

**Discussion**

In 466 - 471: "One of the key findings of our study is the confirmation of the CDA's higher potential over UCDA in reducing the error in both components, thereby legitimising the transition toward WCDA. Our results have implications for NWP, indicating that including the ocean improves the initial state of the atmosphere (Browne et al., 2019). In the case of S2D predictions, where the ocean state is the key source of predictability, this transition — UCDA to WCDA — has already been tested (Balmaseda et al., 2009; Penny and Hamill, 2017; Skachko et al., 2019). In this study, we further present implications for the initialisation of slower modes of variability, showing the importance of atmospheric coupling for such time scales."

2) The observing framework in operational NWP is that of a well-observed Atmosphere and sparsely observed Ocean, which the Authors discuss in Sec. 4.2. This will remain the case for the foreseeable future, even when SWAT altimeters observations become available. For the improvement of weather forecasts in the medium range (1-2 weeks lead time) this is the case that it would be good to see discussed in more detail, possibly in conjunction with a coupled model setup that is realistic for the extra-tropics, i.e. weakly interacting Atmosphere Ocean with the atmosphere dominated by fast, large scale errors and the Ocean with slowly evolving small scale errors.

**R:** Thank you for raising this issue that we did not address clearly in our initial discussions.

We would like first to clarify that, as our model covers several combinations of spatio-temporal scale separations using weak coupling, we could indeed achieve settings that correspond to an 'extratropical' ocean-atmosphere coupling. In fact, our 'standard configuration' used for our system's introduction and description (Sec. 2) with \tau=0.1 and S=1, pertains to "a weak extratropical coupling between the atmosphere and the ocean, …" (Peña and Kalnay, 2004).

On the other hand, we agree with the Reviewer that the well-observed atmosphere scenario is more likely in the immediate future, and that atmospheric initialisation is a key component for skilful NWP, a scenario that we neglected in our discussion. Thus, we addressed this topic in our discussion section as:

In 475 – 440: "... Historically, ocean observations have been scarce compared to the atmospheric network (Laloyaux et al., 2018). We can thus anticipate two situations: first, that the largest benefit of SCDA is expected in the ocean component, meaning that the large-slow ocean modes of variability can benefit from the high-frequency atmospheric variability, potentially improving seasonal-to-decadal predictions, as shown in Sandery et al. (2020). Secondly, in the context of NWP, we can infer that WCDA will remain the best strategy for initializing the atmospheric state in medium-range weather forecasts over the upcoming years. The atmosphere marginally benefits from the improved ocean — obtained with SCDA of atmospheric observations. This little improvement is more evident with the configurations where a large and fast atmosphere interacts with a small and slow ocean, characteristic of the extratropics, where the impact of SCDA is negligible compared to that of WCDA. However, we acknowledge that there has been recent rapid progress in the ocean observation network..."

**References:**

Peña, M. and Kalnay, E.: Separating fast and slow modes in coupled chaotic systems, Nonlin. Processes Geophys., 11, 319–327, https://doi.org/10.5194/npg-11-319-2004, 2004.